# Patent Value and Survival of Patents

**Jung-Tae Hwang [1], Byung-Keun Kim [2,\*] and Eui-Seob Jeong [3]**

[1] Department of Business Administration, Hallym University, Chuncheon 24252, Korea; jthwang@hallym.ac.kr
[2] School of Industrial Management, Korea University of Technology and Education, Cheonan 31253, Korea
[3] Seoul Capital Areal Area Branch, Korea Institute of Science and Technology Information, 66 Hoegi-ro, Dongdaemun-gu, Seoul 02456, Korea; esjng@kisti.re.kr
\* Correspondence: b.kim@koreatech.ac.kr

**Abstract:** This study investigated the effect of patent value on the renewal (survival) of patents. The private value of patents can be one of the main pillars sustaining a firm's value, and the estimation of the value may contribute to the strategic management of firms. The current study aimed to confirm the recent research findings with survival analysis, focusing on the more homogeneous patent data samples. In this study, a dataset is constructed from a cohort of 6646 patents from the 1996 and 1997 application years, using patent data from the European Patent Office (EPO). We found that the family size and non-patent backward citations exhibited profound impacts on patent survival. This result is in line with numerous studies, indicating the positive impact of science linkages in the biotechnology and pharmaceutical fields. It was also found that the effect of the ex-post indicator is not as strong as the ex-ante indicators, like traditional family size and backward citations. In short, the family size matters most for the survival of patents, according to the current research.

**Keywords:** patent renewal; ex-ante indicators; ex-post indicators; survival analysis; Cox hazard regression





## 1. Introduction

The value of a patent to its owners depends on its legal protection of the power of invention, and it is almost analogous to the maximum legal cost—how much they will pay for claiming ownership. Measuring patent value has attracted academic attention [1,2]. A patent portfolio can be critical in appraising the value of a firm [3], the value of a specific research project [4,5] and the performance of a district [6], but a direct evaluation of the asset value of patents is extremely difficult because they are rarely traded. In addition, the formal protection is limited for a designated period (for up to twenty years after filing the application forms), and the span can be from as short as a few years to as long as twenty years, the maximum, depending on the owners' decision to renew the patent. Some researchers have suggested that the changing value of a patent for its lifetime period is due to the obsolescence of the technology, the decreasing protection time remaining, or "invent around"-type competing patents [7]. A recent study on patent renewal fees (renewal fees for the EPO changed from the 3rd year, at 490 euros, to the 10th year and onward, up to 1640 euros. The research used these values for probit regression) showed that the peak for patent value was around 15–16 years after its application [8].

In fact, one critical method to estimate the value is by utilizing the owners' decision to renew the patent at a given fee [7,9]. Because a direct evaluation of the value is difficult and rare (an exceptional case is a study done by Fisher & Leidinger [10], using an Ocean Tomo patent auction), indirect models derived from regressions on renewal records could provide alternative means. In other words, certain critical indicators of a patent could be used in a model to estimate the patent value. Several indicators reflect the value of a patent at the moment it is issued. These ex-ante indicators, such as backward citations of patents and non-patent documents, patent family size, the number of claims, etc., can be incorporated

into a pre-constructed model. At the first decision time for renewal (this is after three years in the case of the EPO), some forward citations may occur and can be considered as an important indicator [1]. These ex-post indicators (either the annual increment of forward citations or the accumulated number of forward citations for several years) may provide valuable information on the patent value. However, which indicators are important and how much weight each one should be given are questions without unanimous opinions. These questions will be addressed in this study.

This research aims at investigating which indicator is the most critical to the survival of a patent—the renewal decision of that patent. Researchers present different models and methods for measuring patent quality, and thus, exhibit inconsistent results. Although citation seems to be the most prominent proxy indicator of patent value in survey-based research [11], other research indicates that the long life span of a patent is highly correlated with the scope (family size) of patents [12]. This research tests a survival model using the cohorts from the application years of 1996 and 1997, as extracted from the patent dataset provided by previous research [1]. The result of the previous research [1] was not based on actual renewal fees but was instead based on the reference renewal fees of patents waiting for the EPO's approval, and this study attempts to adopt an alternative method that does not require the almost unobtainable actual renewal fee figures. It is questionable whether the same result can be achieved if a different method, like Cox hazard regression that emphasizes the life span of patents, is applied to the dataset.

This research contributes to the patent value literature by deepening our understanding of how to measure patent value, through applying the survival analysis method.

## 2. Literature Review and Patent Survival

### 2.1. Review and Critical Patent Indicators

Patent values can be examined in various dimensions, for instance, private value as against public value. Even when discussing private value, the focus may differ; economic value vs. technological value [2,13]. Patents are a major tool to protect intellectual property rights. A patent could be used for defending or blocking the possible entry of competitors, for implementing development by the owner which leads to commercialization, or licensing out so that the appropriate licensee may develop the product and pays out of loyalty to the owner of patent [14]. It is argued that commercialized and licensed patents live longer [15], which implies that a vested interest and partner network indirectly enhance the value of a patent. A patent portfolio is now easily regarded as a reputation signal when firms build strategic alliances of research and development. It is also reported that the litigation to nullify the patent, and confirming the validity of the said patent (survive against the opposition), proves it to be superior to dormant patents [2]. Patent value, as a set of interrelated patents, is supposed to be more efficient for protecting intellectual property rights.

As patent value attracts the interests of academics and industries, measuring this value through different methods is subjected to intense scrutiny. Mutually agreeable tools to measure the value of a patent portfolio may reduce the transaction cost when building a strategic alliance of technological collaboration. This study focuses on a traditional econometric method that consists of patent indicators, to stimulate further discussion on the concept of value. There are several candidates for the dependent variables of proxy values. As Van Zeebroeck [12] identifies, patents that are issued (approved), with litigation records, applied for in triads (the US, the EU, and Japan), and with repetitive renewals, have greater value. The renewal status is of particular concern for the recent measurements, where it is assumed that the renewal fee constitutes the lower limit of the patent private value [1]. In addition, renewal status does not only reveal the snapshot value of a patent. If the data are accumulated over the life span of patents (twenty years of protection), the renewal status may provide opportunities to apply survival analysis.

Numerous studies have adopted econometric methods, including probit, the Cox hazard function, or both of them [16] to determine which patent indicators are more

influential in determining patent value with regard to patent renewal. Unlike a previous study that used the same EPO pharmaceutical patent database [1], this study adopted survival analysis with the Cox time-varying covariates hazard function. The reason for using the Cox model was to consider which indicators are more influential on repeated decisions to renew, thus causing a longer life span for a patent.

The inclusion of patent indicators varies. In the case of utilizing ex-ante indicators, Harhoff et al. [17] even used renewal records of backward citation patents (those cited patents). Some researchers also included backward citation lag [18], the scope of the backward citations as originality [19], and the scope of the patent as the number of 4-digit IPC subclasses [16]. A recent study categorized forward citations into two groups, and argued that the incremental forward citation (added number of forward citations) is one of the most critical indicators [1]. One study showed that the relationship between citations and patent value was log-linear [20].

To manage difficulties that occur in evaluating patent value, researchers chose to focus on certain critical indicators that define patent quality. In fact, Lanjouw and Schankerman [3] noted that the use of multiple indicators contributes to the reduction in quality variance, thus leading to robust evaluation. In the current research, we included ex-post indicators and ex-ante indicators following the recent work on EPO patents [1]. In various studies [16], the number of forward citations has been noted as an "ex-post" indicator, but its use includes the five-year accumulated forward citations in general. In this study, we adopted the accumulated number over twenty years in each survival.

### 2.2. Hypothesis

Van Zeebroeck [16] indicated that the number of five-year forward citations has a greater influence than the number of backward citations, in his investigation into the massive sample of the EPO patent dataset. Family size has been highlighted in several studies as a very important cursor to reveal the market value of a patent [2,16]. Based on the multitude of studies that support the positive and significant influence of patent family size on patent value [2,10,21–26], we might postulate that the size of the patent family is the most significant indicator. However, whether ex-ante indicators are more significant than an ex-post indicator is not certain, due to a recent research finding [1]. Because Fisher and Leidinger [10] emphasized the impact of family size more than that of forward citations, it is worthwhile to re-examine their previous research findings.

Research hypothesis: the ex-post indicator (i.e., forward citation) is a more significant influencer than ex-ante indicators for patent survival (patent quality).

Here, we need to emphasize that patent survival does not equate patent value. The life span of a patent only captures one facet of private value. The hypothesis is equivalent to the following research question: "Does forward citation significantly affect the survival of a patent?"

## 3. Method and Variables

### 3.1. An Analytical Model for Patent Survival (or Failure: Non-Renewal)

The conceptual model of the current study is simple. A higher private value of a patent leads to a longer patent life (repetitive survival), which leads us in search of an appropriate survival model. In this regard, it is important to illuminate which indicator is critical for the longer life span of patents.

This study adopted a time-dependent Cox regression survival analysis, based on two cohorts of patents applied for in 1996 and 1997. One of the reasons to pick only two cohorts was to secure a long life span of 17 and 16 years (considering the first renewal comes three years after application). The original database used in the previous study contains cohorts for the years 1996–2009, but we intentionally selected cohorts with longer life spans to obtain better results from the survival analysis (Cox hazard) model.

We considered time-varying explanatory variables associated with event history and censoring, because we utilized a longitudinal dataset [27]. This study used right censoring:

the patents were censored when the patents faced the end of the observation period (the year 2016) before reaching their full life span. In fact, this is only applicable to the 1997 cohort. The other problem is the continuously changing explanatory variables over time. To address these two problems, we used a Cox regression with time-varying covariates.

In a Cox hazard model, the hazard function represents the probability that a failure occurs at time t. It is assumed that an individual (patent) does not experience a failure (non-renewal) prior to T (random time-period variable of the event). Following the formula suggested by Allison (1982) [27], the following log-linear equation represents the proportional hazard function:

$$log\lambda(t,x) = \alpha(t) + \beta x + \beta^{\mathsf{T}} x(t) \tag{1}$$

$$\lambda\left\{(t|X,\ X(t)) = \lim_{h\to0+}\left\{\frac{Pr(t \le T \le t+h|T \ge t, X, X(t))}{h}\right\} = \lambda_0(t)\exp(\beta X + \beta^{\mathsf{T}} X(t)) \tag{2}$$

where $\alpha(t)$ is an unspecified baseline hazard; $\beta$ is the coefficient vector of the explanatory $\chi$ variables and $\beta^{\mathsf{T}}$ is the coefficient vector of the time-varying explanatory $\chi(t)$ variables on the instantaneous probability of an event. Equation (2) shows that the hazard of a failure is related to the covariates [28,29]. Furthermore, as we have seen in the equation, the proportional hazard model can be easily extended into a time-varying coefficient [29].

### 3.2. Patent Value Indicators

Here, five ex-ante indicators of patent family size, patent backward citations, non-patent (literature) backward citations, number of claims, and number of inventors, are presented. In addition, one ex-post indicator, forward patent citations (received from patents applied for later) is tested. All indicators were pseudo-normalized to reduce skew, by taking the log (1+ indicator) and then linear scaling those (min, max) values into values between zero and one [0, 1].

### 3.2.1. Patent Family Size

The family size portrays the geo-demographic scope of the market protection of a patent. It is defined as the number of countries where a given invention is filed. After the pioneering research of Putnam [21], a multitude of researchers confirmed the high significance of this indicator and its irrefutable correlation with patent value and patent quality [10,22–26].

This ex-ante indicator provides more economic information that is pertinent to the value of a patent, such as market sizes or potential market structure. As such, we expected that our study would also show the importance of this indicator. Following the previous study [1], we computed the indicator (FAT) from the PATSTAT Biblio database provided by the EPO.

### 3.2.2. Patent Backward Citations

Patent backward citations indicate how many citations were assigned to patents applied for earlier. Some scholars insist that patent backward citations quantify the breadth (scope) of a patent and implicate the technological novelty of the invention [19,30,31]. Much literature has shown the significant and positive impact of this citation indicator on the patent value [2,7]. We computed the patent backward citations (PBKWCIT) from the PATSTAT Biblio database.

### 3.2.3. Non-Patent Backward Citations

This indicator is similar to PBKWCIT but refers to the number of references made to non-patent literature instead of to earlier patents. In various previous studies, this indicator has been featured as the science linkage; it portrays the relationship of a patent to the academic literature [18,32]. The significance of non-patent backward citations may depend on the technology fields, and researchers found that this indicator is particularly prominent for chemical and pharmaceutical patents [2]. We computed this indicator (NPBKWCIT) utilizing the PATSTAT Biblio database.

### 3.2.4. Number of Claims

This indicator is supposed to reveal the scope of legal protection for a patent [22]. Some researchers, however, have argued that it could be misleading because different institutional systems cause a significant difference in the number of claims [33]. Nonetheless, within the same institutional setting, patents with numerous claims are supposed to have a higher likelihood of renewal and

higher values. We computed the number of claims (CLAIMS) utilizing the EPO's PATSTAT Biblio database.

### 3.2.5. Number of Inventors

This indicator provides information on the size of the research team [25]. It is debatable whether the number of inventors is positively correlated with the patent value. Due to the conflicting findings of previous studies [34,35], we tested the associations between the number of inventors and patent survival. A large research team size (i.e., a large number of inventors) may often be observed in academic research, and it would be constructive to include a control variable for the type of assignees. However, the current dataset does not contain information on the organization type; thus, we only considered the number of inventors (INV). We computed this indicator from the PATSTAT Biblio database.

### 3.2.6. Forward Citations

Unlike other ex-ante indicators, forward citations are the ex-post indicator in this study that best describes the influence of the patent on future inventions. The number of citations received from later patents has been studied as possible measures of values such as social values [36,37] and private values of patents [2,10]. However, some take care to use forward citations as the best ex-post indicator, because the relationship between forward citations and patent values is not clear [30]. In addition, using forward citations requires the introduction of an additional control variable regarding technology fields to reduce the sector bias [24]. Because we restricted the scope of the patent dataset to only the pharmaceutical field, we skipped this control variable.

In this study, two measures of forward citations were adopted. The first, to estimate the overall impact of forward citations, the total number of forward citations until a specific time (i.e., year t) (ACCFWDCIT, "accumulated citations") was used. The second, to estimate the marginal effects of forward citations, the number of forward citations at the time span (i.e., from year t-1 to year t) (INCFWDCIT, "incremental citations") was used. The forward citations is computed from the EPO's PATSTAT Biblio database.

## 4. Data

Dual databases provided by the EPO were used in this study: the PATSTAT Biblio database and the PATSTAT Legal Status database. With coding of Structured Query Language (SQL), the targeted patent-related details were extracted. The method to extract the dataset followed the previous research [1], which used international patent classification relevant to pharmaceutical fields, as designated by Schmoch's 2008 *Report to the World Intellectual Property Organization*. The core list of patents was collected from the PATSTAT Biblio database (2016 Autumn Edition), on the condition that the application was relevant to an existing granted patent and was filed at the EPO. In addition, by querying for specific IPCs, the technology field is restricted to the pharmaceutical sector. For each patent, five ex-ante indicators, patent family size, number of inventors, patent backward citations, non-patent backward citations, number of claims, and one ex-post indicator, forward citations, were constructed by querying the PATSTAT Biblio database. The collection of patent renewal statuses for the patent cohorts of the application filing year, by querying both the PATSTAT Biblio database and the PATSTAT Legal Status database (2016 Autumn Edition), yielded 34,106 patents [1].

In this study, the applications (which was recalculated from the last renewal year and age of the patent) of the year 1996 and 1997 cohorts were extracted from the 34,106 patents. The total number of those cohorts was 6634. Table 1 presents the descriptive statics.

**Table 1.** Descriptive statistics of patent indicators.

|  | N | Min | Max | Mean | SD |
|---|---|---|---|---|---|
| FAT | 6646 | 1 | 107 | 17.09 | 11.458 |
| INV | 6646 | 1 | 22 | 3.48 | 2.384 |
| CLAIMS | 6646 | 1 | 135 | 15.89 | 11.496 |
| PBKWCIT | 6646 | 0 | 10 | 0.11 | 0.547 |
| NPBKWCIT | 6646 | 0 | 33 | 0.20 | 1.004 |
| INCFWDCIT _y19 | 6646 | 0 | 7 | 0.02 | 0.189 |
| ACCFWDCIT _y19 | 6646 | 0 | 41 | 0.29 | 1.432 |

As for the time-count reference in the survival model, a patent age from 1 to 20 constitutes a discrete time-lapse of the Cox regression in this study. However, the effective time span for the 1996 cohorts was only seventeen years of discrete time, because the renewal decision under the European Patent Convention is taken between the third and nineteenth year after filing patents (17 renewal decisions). Furthermore, we post-processed the 1996 cohort's final year renewal as right-censored in the Cox regression (renewal decision 16 times) to enhance the homogeneity with the 1997 cohort.

## 5. Results

Table 2 shows that the B (hazard coefficient of covariates) value is positive for the number of inventors (INV), which means having many inventors on a patent increases the risk of demise (non-renewal of the patent). This finding is in agreement with a previous study [1]. Between the two derived ex-post indicators, the traditional accumulated forward citations (ACCFWDCIT) are significant. In all three models, incremental forward citations are not statistically significant, and neither are backward patent citations. Regarding citations, only backward non-patent literature citations and accumulated forward citations positively influence patent survival (negatively affecting hazard).

**Table 2.** Cox regression result.

|  | Model 1 | | Model 2 | | Model 3 | |
|---|---|---|---|---|---|---|
|  | **B** | **P sig.** | **B** | **P sig.** | **B** | **P sig.** |
| CLAIMS | −0.093 | 0.280 | −0.098 | 0.252 | −0.093 | 0.279 |
| INV | 0.243 | 0.001 | 0.243 | 0.001 | 0.243 | 0.001 |
| FAT | −0.726 | 0.000 | −0.732 | 0.000 | −0.726 | 0.000 |
| PBKWCIT | −0.135 | 0.301 | −0.139 | 0.289 | −0.135 | 0.302 |
| NPBKWCIT | −0.550 | 0.000 | −0.548 | 0.000 | −0.550 | 0.000 |
| INCFWDCIT | −0.105 | 0.706 | −0.345 | 0.140 |  |  |
| ACCFWDCIT | −0.293 | 0.114 |  |  | −0.332 | 0.033 |
| Model fit. $\chi^2$ | 100.970 | (7) | 98.654 | (6) | 100.841 | (6) |

Table 2 presents the results of the Cox regression with the time-varying covariates.

The result in Table 2 shows the outright importance of "patent family size" as an indicator and reveals that backward non-patent citations outweigh forward citations. This finding contradicts the argument of a recent previous study that the increment of forward citations is more critical to patent value [1]. The overall fit of the models also indicates that accumulated forward citations have a greater explanatory value (Model 3 of Table 2).

Recalling the research hypothesis on the importance of an ex-post indicator, we found that family size is the most important indicator for the survival of the current 1996–1997 cohorts. Considering the amount that is invested in a patent family (applying for patents in a multitude of countries), the additional renewal fees could be less important for the owner, and the *hitherto* investment may have contributed to the survival of patents. Therefore, ex-post (forward citations) does not represent the most significant indicator for patent survival, unlike in the previous study.

## 6. Discussion

In the pursuit of a better understanding of the role of ex-ante and ex-post indicators for a patent's survival, we adopted an alternative method for evaluating the importance of patent indicators. We thought that the renewal fee is not important for the same cohort of patents when implementing the Cox proportional hazard model, but the result is quite different. The research question was answered that the ex-post indicator (forward citation) is not as influential as the ex-ante family size indicator for patent survival.

For this discussion, we consider the seven indicators in Table 2 one by one. The number of claims (CLAIM) is supposed to be important according to former research [22], but this ex-ante indicator is found to not be significant in the current research. We presume that other variables like patent family size may slightly be correlated and represent the possible potential applications, undercutting the importance of CLAIM. The number of inventors (INV) negatively affects patent survival. This result does not surprise us, because both positive and negative effects were reported in previous research [34,35]. The size of the patent family (FAT), as explained above, represents the most important and significant factor, positively affecting patent survival. Unlike forward citation,

this is more closely related to the potential market size than to scientific value. The number of cited previous patents, backward citation (PBKWCIT), is not significant, and we suspect that the field of research–biotechnology, with emphasis rather on the linkage to science than to technology, may result in the non-significance of PBKWCIT. Therefore, highlighting the linkage to science, the number of non-patent backward citations (NPBKWCIT) usually citing scientific papers stands out in the current study. It is the second most significant indicator in Table 2 positively affecting the survival of patents. The ex-post indicators that represent the scientific and technical potentials of a patent, incremental forward citations (INCFWDCIT), and accumulated forward citations (ACCFWDCIT), were expected to be significant. However, it was found that only ACCFWDCIT is statistically significant, at 5%. This is different from the previous research [1] and appears to be mysterious, which we will discuss further in the next paragraphs.

In this study, the standardizing method z-score was not used. Instead, this study used log transformation and rescaled the covariates into the [0, 1] range, which might also make the result different from the previous study. In addition, the previous study [1] used renewal fees set by and paid to EPO, which vary significantly along with the time lapse [1]. However, the previous study did not consider the actual renewal fees (possibly higher if target countries for protection are many) in member countries. This study utilized simple annual survival (renewal) status, and it did not consider the uneven financial load of renewal.

In fact, both methods may have defects, because the renewal decision depends on "private value," and the decision at the end of a patent's life span is influenced by the owner's resources and patenting policy. It is very common for a firm to decide not to renew patents in the final year of protection, because of no future protection (low optional value) twinned with expensive renewal fees. The renewal decision is also vulnerable to a competitive landscape. For example, the renewal decision may be switched suddenly in the situation when a rival technology patent has a strong market position, and its inherent or potential academic values may not be well appreciated at the time of the renewal decision. This may confuse the regression analysis, as the combination of indicators tries to capture the whole value. Technological value attracts continuous future citations, but it does not translate into renewal value when the corresponding patent has lost competition in the market. It is paradoxical that regression analysis is more appropriate for finding the general value of a patent, but the renewal decision is prone to capricious private economic value. In this sense, Harhoff and his colleagues' argument on the difference between "renewal value" and "asset value" [2] is highly relevant.

On second thought, the technological value, by enhancing the image of the firm as a pioneer and technology leader, may provide a certain private value just as a "commercial brand". Scientific advancement is the most well-known public value of a patent [38], and the reputation aspect of value is ambidextrous, encompassing both private and public value. Since any discussion on patent value is critically related to building a sustainable firm strategy to manage intellectual property and strategic alliance of technological collaboration, it is necessary to dictate the implications of measurement tools. The current study suggests that even though the tools were built to measure private value (and coefficients extracted from regression on renewal data), the model itself may reflect wider aspects of value, and consequently be useful for estimating potential research partners.

For future research directions, it is notable that recent developments in artificial intelligence may contribute to calculating patent values from the actual renewal fees of patents. In this case, the previous model [1] may be retrieved as its formulation incorporates renewal fees.

## 7. Conclusions

The current research reconfirms that both ex-ante indicators and the ex-post indicator are influential factors for patent survival. Regarding the research question, the most significant factor that decides the life span of a patent is the family size of a patent. This result is in agreement with the findings of Fisher and Leidinger [10], but differs from a previous study [1]. The previous finding on the strong correlation between the life span of a patent and family size [39] seems to be one explanation why the results deviated from the former research [1].

The results of this study contribute to a prolonged line of research—measuring patent value. However, the results also presented an implication conflicting with a previous study [1]. Both studies adopted an analysis based on renewal decisions, and this study simply used repeated renewal decisions, unlike the previous study, which used renewal fees [1]. Even though this study used a simple method with a more homogeneous dataset, it does not mean the analytical results are more reliable, which opens further discussion on methodological aspects. As Harhoff et al. [2] indicated,

the best solution is not to rely on a sole indicator but to resort to the combination of available information.

The limit of the previous research [1] still remains in the current research. The renewal status constitutes the minimum protection value for the patent owners. The assumption that the owners acknowledge the intrinsic value of the patent with perfect information at the time of renewal decision is a very strong condition. Without an auction record of patents, which rarely happens, the proxy value is hard to obtain.

Finally, we have to confess another limitation of this research, namely, that it does not incorporate the recent development of patent value estimation. The development of computer science, big data processing, and artificial intelligence is now advancing to incorporate deep learning and text-mining of whole patent documents [40,41]. In the future, a research project that simultaneously incorporates expert monetary evaluation on a large sample of patents, text-mining and artificial intelligence evaluations of patent value, and a traditional evaluation of patent indicators (regression) may shed light on a more comprehensive estimation of patent values.

Analyzing different sets of patent data may lead to comparable results. We used the pharmaceutical technology from the European Patent Office (EPO). This can be extended to other technology fields, to compare what indicators influence patent values between technology fields. If researchers construct a patent database that includes the U.S., Japan, and other countries, it may provide opportunities to find differences in the effects of patent policies on patent value between countries.

**Author Contributions:** Formal analysis, J.-T.H.; funding acquisition, B.-K.K.; investigation, J.-T.H.; methodology, J.-T.H. and E.-S.J.; project administration, B.-K.K.; resources, E.-S.J.; validation, E.-S.J.; writing—original draft, J.-T.H.; writing—review & editing, B.-K.K. All authors have read and agreed to the published version of the manuscript.

**Funding:** This work was supported by the Ministry of Education of the Republic of Korea and the National Research Foundation of Korea (NRF-2019S1A5C2A02082342).

**Data Availability Statement:** Publicly available datasets were analyzed in this study. This data can be found here (https://www.epo.org/searching-for-patents/data/web-services.html, accessed on 25 April 2021).

**Conflicts of Interest:** The authors declare no conflict of interest.

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
