# Peer review of "Patent Value and Survival of Patents"

_2199-8531, doi:10.3390/joitmc7020119_

Round 1
Reviewer 1 Report
The paper lacks a clear motivation, and the introduction section is rather lacking motivation and not well structured. Authors need to highlight their contributions and mention that which part of their results are new and fill the research gaps. Also, there is a need to state the method use and data sources in the introduction.
The introductory discussions that precede each of the formulated hypotheses are insufficient and more background studies should be studied in support of these hypotheses.
In data section there is no information about which processing took place. Rewriting the data section in a way that other researchers can replicate the data collection and analysis is necessary.
The empirical approach itself is not sufficient. Moreover, the authors do not provide any useful robustness test, they neither augment their dependent variables (for example different citation window) nor do they put any variation on the independent or control variables.
I found the discussions of the findings inadequate.
The conclusion is basically a repeat of the discussion and besides does not add much to the final results.
Author Response
Thank you very much for the detailed feedback on our submission. Please see the attachment.

Reviewer 2 Report
The topic of the paper is relevant and current, as it represents a competitive advantage for companies, namely in the area of biotechnology and pharmaceutical products.
The Abstract is well prepared and has all the essential contents of the investigation.
The literature review is sufficient and appropriate to answer the research question. The authors should have formulated hypotheses during or at the end of the literature review.
The statistical methods used are suitable for processing the data and answering the research question.
In general, the paper is well structured; there is a coherence between the topic - research question - and the scientific methodology followed; the conclusions identify the most relevant results achieved; the references are current.
Suggestions for authors:
- Formulate a conceptual model
- Formulate hypotheses
Author Response
Dear Reviewers:
Thank you very much for the detailed feedback on our submission and for the opportunity to revise and resubmit our manuscript to Journal of Open Innovation: Technology, Market, and Complexity. We would also like to express our sincere gratitude for your insightful comments. We have worked hard to revise the manuscript by carefully implementing the suggestions made by the reviewers. The revisions are described in the remainder of this response document.
We hope that we have addressed all the issues raised to your satisfaction. We will be happy to make any additional changes you deem necessary. We thank you for giving us the opportunity to improve our paper.
- Formulate a conceptual model
Thank you for suggesting possible strategies for improvements of this research. The conceptual model of current study is simple. The higher private value of a patent leads to longer patent life (repetitive survival), which leads us in search of appropriate survival model. In this regards, it is important to illuminate which indicator is critical for longer life span of patents.
- Formulate hypotheses
Thank you for suggesting possible strategies for improvements of this research. Research hypothesis: Ex-post indicator (i.e., forward citation) is a more significant influencer than ex-ante indicators for patent survival (patent quality).
Reviewer 3 Report
The article entitled Patent Value and survival of Patents seems to be interesting. I have a few comments:
The Introduction must be improved with the inclusion of the aim of the study. After the presentation of the relevance of the research topic, the aim of the study should be formulated (clearly and specifically) in the introduction.
Keywords should reflect the main idea and content of the article. The purpose of keywords is to provide the insight to the reader into the contents of the paper. They should reflect the area of the research with no need to replicate words from the title of the manuscript.
The text of the article should not be submitted in the first person (personified). Personified - it means not "we" and not "authors". It means in the passive voice, for example-The paper ex
In page 6 the word hithertoi is in red, why?
A section or paragraph “Directions for future research” should be added before conclusion.
On the basis of detailed comments sent to the authors, I porpose a minor revision of the article prior to its publication.
Author Response
Dear Reviewers:
Thank you very much for the detailed feedback on our submission and for the opportunity to revise and resubmit our manuscript. Please see the attachment.

Round 2
Reviewer 1 Report
The paper has been improved but authors need to do a thorough literature review to support their hypotheses. The background studies are insufficient in the current version.
The conclusion section itself is not well-written. It is just a repetition of their results. That would be good if authors add more discussions to this part, for example, what do we learn from these findings, what are the policy implications? how and for whom these findings would be useful? etc.
Author Response
2nd Revision Report
Patent Value and survival of Patents
Overall Summary of Modifications
Dear Editor and Reviewers:
Thank you very much for the detailed feedback on our submission and for the opportunity to revise and resubmit our manuscript to Journal of Open Innovation: Technology, Market, and Complexity. We would also like to express our sincere gratitude for your insightful comments. We have worked hard to revise the manuscript by carefully implementing the suggestions made by the reviewers. The revisions are described in the remainder of this response document.
We hope that we have addressed all the issues raised to your satisfaction. We will be happy to make any additional changes you deem necessary. We thank you for giving us the opportunity to improve our paper.
Reviewer 1
Point 1. The paper has been improved but authors need to do a thorough literature review to support their hypotheses. The background studies are insufficient in the current version.
Thank you for suggesting possible strategies for improvements of this research. We revised literature review.
There are several candidates for the dependent variables of proxy values. As Van Zeebroeck [12] identifies, patents that are issued (approved), with litigation records, applied in triads (the US, the EU, and Japan), and with repetitive renewals have greater value. The renewal status is of particular concern for the recent measurements, it is assumed that the renewal fee constitutes the lower limit of the patent private value [1]. In addition, renewal status reveals not only the snapshot value of patent. If the data is accumulated for the life span of patents (twenty years of protection), the renewal status may provide opportunities to apply survival analysis.
To manage difficulties that occur in evaluating patent value, researchers choose to focus on some critical indicators that define patent quality. In fact, Lanjouw and Schankerman [3] noted that the use of multiple indicators contributes to the reduction of quality variance, thus leads to robust evaluation.
Point 2. The conclusion section itself is not well-written. It is just a repetition of their results. That would be good if authors add more discussions to this part, for example, what do we learn from these findings, what are the policy implications? how and for whom these findings would be useful? etc.
Thank you for suggesting possible strategies for improvements of this research. We revised conclusion.
The limit of previous research [1] still remains in the current research. The renewal status constitutes the minimum protection value for the patent owners. The assumption that the owners acknowledge the intrinsic value of patent with perfect information at the time of renewal decision, is very strong condition. Without auction record of patents, which rarely happens, the proxy value is hardly obtainable.
Analyzing different set of patent data may lead to comparable results. We used the pharmaceutical technology from the European Patent Office (EPO). It can be extended to other technology fields to compare what indicators influence patent values between technology fields. If researchers construct patent database including the U.S., Japan, and other countries it may provide opportunities to find differences in the effects of patent policies on patent value between countries.

Round 3
Reviewer 1 Report
The paper is significantly improved in the revised version.